# Concurrent Chemoradiation in Anal Cancer Patients Delivered with Bone Marrow-Sparing IMRT: Final Results of a Prospective Phase II Trial

**DOI:** 10.3390/jpm11050427

**Published:** 2021-05-18

**Authors:** Francesca Arcadipane, Patrick Silvetti, Francesco Olivero, Alessio Gastino, Roberta Carlevato, Ilaria Chiovatero, Lavinia Spinelli, Massimiliano Mistrangelo, Paola Cassoni, Giuliana Ritorto, Elena Gallio, Adriana Lesca, Riccardo Faletti, Francesca Romana Giglioli, Christian Fiandra, Umberto Ricardi, Pierfrancesco Franco

**Affiliations:** 1Department of Oncology, Radiation Oncology, AOU Citta’ della Salute e della Scienza, 10126 Turin, Italy; francesca.arcadipane@gmail.com; 2Department of Oncology, Radiation Oncology, University of Turin, 10126 Turin, Italy; patrick.silvetti@gmail.com (P.S.); francesco.olivero668@edu.unito.it (F.O.); alessio.gastino@gmail.com (A.G.); roberta.carlevato@gmail.com (R.C.); ilakiowa@hotmail.it (I.C.); lavinia.spinelli1@gmail.com (L.S.); christian.fiandra@unito.it (C.F.); umberto.ricardi@unito.it (U.R.); 3Department of Surgical Sciences, Abdominal Surgery, University of Turin, 10126 Turin, Italy; mistrangelo@katamail.com; 4Department of Medical Sciences, Pathology Unit, University of Turin, 10126 Turin, Italy; paola.cassoni@unito.it; 5Department of Oncology, Oncological Centre for Gastrointestinal Neoplasm, AOU Città della Salute e della Scienza, 10126 Turin, Italy; giuliana.ritorto@libero.it; 6Medical Physics Unit, S.C. Fisica Sanitaria, A.O.U. Città della Salute e della Scienza, 10126 Turin, Italy; gallio.elena@gmail.com (E.G.); fgiglioli@cittadellasalute.to.it (F.R.G.); 7Division of Nuclear Medicine, AOU Città della Salute e della Scienza, 10126 Turin, Italy; alesca@cittadellasalute.to.it; 8Department of Surgical Sciences, University of Turin, 10126 Turin, Italy; riccardo.faletti@unito.it

**Keywords:** anal cancer, bone marrow-sparing IMRT, hematologic toxicity, radiotherapy

## Abstract

We investigated the role of the selective avoidance of haematopoietically active pelvic bone marrow (BM), with a targeted intensity-modulated radiotherapy (IMRT) approach, to reduce acute hematologic toxicity (HT) in anal cancer patients undergoing concurrent chemo-radiation. We designed a one-armed two-stage Simon’s design study to test the hypothesis that BM-sparing IMRT would improve by 20% the rate of G0–G2 (vs. G3–G4) HT, from 42% of RTOG 0529 historical data to 62% (α = 0.05; β = 0.20). A minimum of 21/39 (54%) with G0–G2 toxicity represented the threshold for the fulfilment of the criteria to define this approach as ‘promising’. We employed ^18^FDG-PET to identify active BM within the pelvis. Acute HT was assessed via weekly blood counts and scored as per the Common Toxicity Criteria for Adverse Effects version 4.0. From December 2017 to October 2020, we enrolled 39 patients. Maximum observed acute HT comprised 20% rate of ≥G3 leukopenia and 11% rate of ≥G3 thrombocytopenia. Overall, 11 out of 39 treated patients (28%) experienced ≥G3 acute HT. Conversely, in 28 patients (72%) G0–G2 HT events were observed, above the threshold set. Hence, ^18^FDG-PET-guided BM-sparing IMRT was able to reduce acute HT in this clinical setting.

## 1. Introduction

Combination therapy with concurrent chemo-radiation (RT-CHT) is considered standard of care for the curative treatment of patients with epidermoid carcinoma of the anus, with a high chance of tumor control and patient survival, accompanied by anatomical and functional preservation of the anal sphincter complex [1,2]. Compliance to therapy is crucial in this setting to maintain the pre-established treatment intensity, given that unscheduled treatment breaks and consequent increased overall treatment may be detrimental on clinical outcomes [3]. Minimizing the toxicity profile during RT-CHT in anal cancer is hence of paramount importance, particularly to limit major events. Among all toxicities, hematologic toxicity (HT) can be critical because it may potentially increase the likelihood to experience infections, bleeding, or anemia-correlated symptoms, eventually hampering treatment compliance [4]. When radiation therapy is delivered with conventional techniques, grade 3–4 HT can be as high as 61%, as shown in the 5-fluorouracil/mytomicin C arm of the RTOG 9811 trial [5]. Intensity-modulated radiotherapy (IMRT), being able to improve dose distribution, with higher conformity and abrupt dose fall off compared to standard techniques, may decrease acute HT, as shown in the RTOG 0529 trial, where the likelihood to experience ≥G2 hematologic events was decreased as compared to historical data [6]. Nevertheless, in the same trial, the observed rate of ≥G3 acute HT was as high as 58%, probably due to the absence of a specific planning strategy targeted at the anatomic structures harboring hematopoietic function [4,6]. Even if chemotherapy is the strongest trigger for HT in this setting, radiotherapy plays an important role, considering that circulating blood cells and precursors within bone marrow (BM) are exquisitely radiosensitive [4]. It was shown in previous studies, that the unintended dose received by the osseous segments within the pelvis, identified either on computed tomography or with ^18^fluorodeoxyglucose (FDG)-labeled positron emission tomography (^18^FDG-PET) to segment hematopoietically active BM, is associated with the rate of occurrence and severity of acute HT in anal cancer patients treated with concomitant RT-CHT [7,8,9,10,11]. Taking advantage of IMRT planning and delivery, active BM comprised within the pelvic region may be used as an organ at risk to be accounted for in the optimization process in order to minimize the dose received by it and consequently spare hematopoietic precursors [12]. We, hence, conducted a single-arm mono-institutional prospective phase II trial, to test the hypothesis that the selective sparing of hematopoietically active BM, as identified with ^18^FDG-PET, may decrease the acute HT profile in anal cancer patients undergoing concurrent RT-CHT. The first phase of the study, previously reported, yielded to promising clinical results, prompting us to conclude the second part of this prospective phase II, whose final results we hereby describe [12].

## 2. Materials and Methods

### 2.1. Eligibility Criteria

Detailed eligibility criteria have been already reported [12]. Briefly, all patients had a histological diagnosis of squamous cell anal carcinoma of the anus and were staged with pelvic magnetic resonance, chest computed tomography and whole body ^18^FDG-PET, according to the Seventh Edition of the American Joint Committee on Cancer staging manual [13]. All cases included were staged as T1-T4, N0-N3 and treated with concurrent RT-CHT with definitive intent. Written informed consent was obtained for all patients. The study was conducted in accordance with the Declaration of Helsinki and the protocol was approved by the Ethics Committee of AOU Citta’ della Salute e della Scienza, Turin, Italy (project identification code: 0089578). The trial was registered in the internal repository for clinical trials at AOU Citta’ della Salute e della Scienza, Turin, Italy (Project identification code: 1190/2016).

### 2.2. Study Design and Sample Size Determination

A one-armed optimal two-stage Simon’s design was selected, to test the hypothesis that treatment modality under investigation (BM-sparing IMRT) would increase by 20% the rate of G0–G2 (vs. G3–G4) acute HT over the historical data observed with the IMRT approach as employed within the RTOG 0529 trial, where the observed rate of ≥G3 acute HT was 58% (rate for G0–G2: 42%) and no targeted optimization toward active BM was used, null hypothesis (H_0_): no difference in acute HT between treatment modalities. [6,14]. The present study was based on the following assumptions: (a) the historical data of success (p0) was represented by the 42% rate of G0–G2 acute skin toxicity (G3–G4:58%) detected in the RTOG 0529 study; (b) the threshold of successful trial (p1) with the treatment schedule under investigation (BM-sparing IMRT) was set to 62% of G0–G2 acute HT (G3–G4: 38%); (c) the α error (one-sided type I error) was set at 5%; (d) the β error at 20% (type II error; power 80%). At the first stage, among 21 enrolled patients, at least 9 (43%) should experience G0–G2 acute HT to further proceed with the trial. In the first step of the trial, we reported a rate of 81% for G0–G2 acute HT (17/21 patients) [12]. At the second stage, another 18 patients were planned to be accrued for an overall sample size of 39 patients. A minimum of 21/39 (54%) with G0–G2 toxicity represented the threshold for the final rejection of H_0_ and the fulfilment of the criteria for the definition of a ‘promising’ treatment for BM-sparing IMRT.

### 2.3. Radiotherapy Protocol

Detailed radiotherapy protocol can be found in Arcadipane et al. [12]. Briefly, the gross tumor volume (GTV) for both primary and nodes was expanded isotropically (20 and 10 mm, respectively) to generate the clinical target volumes (CTVs) and edited to exclude bones and muscles. The elective CTV comprised the mesorectum, pelvic nodes and inguinal groins. Lymphatic areas were contoured as an 8–10 mm isotropic expansion around regional vessels. To generate the corresponding planning target volumes (PTVs), a 10 mm isotropic margin was added to all CTVs [15,16,17].

Dose prescription followed the RTOG 0529 indications, adapted on clinical stage at diagnosis [6]. Patients with cT2N0 disease were administered 50.4 Gy/28 fractions (1.8 Gy daily) to the primary tumor, while elective nodes were prescribed 42 Gy/28 fractions (1.5 Gy/daily). Patients presenting with cT3-T4/N0-N3 disease were prescribed 54 Gy/30 fractions (1.8 Gy daily) to the GTV, while gross nodal disease was prescribed 50.4 Gy/30 fractions (1.68 Gy daily) if sized ≤ 3 cm or 54 Gy/30 fractions (1.8 Gy/daily) if > 3 cm. Elective nodal volume was prescribed 45 Gy/30 fractions (1.5 Gy/daily) [6,18].

### 2.4. Bone Marrow Segmentation

The outer contour of pelvic bone marrow (PBM) was delineated on the planning computed tomography employing a bone window. The PBM was outlined entirely and then divided into three subsites, as first described by Mell et al. to include (a) the iliac BM (IBM), (b) the lower pelvic BM (LPBM) and (c) the lumbosacral BM (LSBM). To segment active BM, a rigid co-registration was perfomed between planning computed tomography and diagnostic ^18^FDG-PET [19]. Thereafter, ^18^FDG-PET standardized uptake values (SUVs) were calculated for PBM volumes, after correcting for body weight. Active bone marrow (^ACT^BM) was defined as the volume having higher SUV values than the SUV_mean_ for each patient, as proposed by Rose et al. [10]. Those areas segmented within PBM were named ^ACT^PBM and those comprised within the three subregions named ^ACT^LSBM, ^ACT^IBM, and ^ACT^LPBM, respectively. Inactive BM (^INACT^BM) was outlined as the difference between PBM and ^ACT^PBM and then segmented for each of the three subregions. Details on the segmentation process can be found elsewhere [9,11,12].

### 2.5. Planning Process and Delivery

All treatment plans were generated using the Monaco treatment planning system version 5.11 (Elekta, Stockholm, Sweden) or Raystation version 10A (Raysearch, Laboratories AB, Stockholm, Sweden), optimization on both PTV and organs at risk with biological cost-functions. A volumetric modulated arc therapy (VMAT) approach was used to combine a rotational geometry, beam modulation obtained by continuous modulation of multileaf collimator, dose rate variations and gantry rotational speed dynamics [15]. The optimization strategy addressed priorities to both target volumes and organs at risk (bladder, external genitalia, large and small bowel, and femural heads), dose fall-off, maximum dose, and cold spot management. The full set of dose constraints for active pelvic BM targeted both PBM and LSBM and included ^ACT^PBM V_10_ < 90%, ^ACT^PBM V_20_ < 75%, ^ACT^LSBM V_40_ < 41%, ^ACT^LSBM mean dose < 32 Gy [9,11,12,20]. The treatment was finally delivered using the Elekta Synergy platform.

### 2.6. Chemotherapy

Patients received concurrent chemotherapy, consisting of 5-fluorouracil (1000 mg/m^2^/day) administered as continuous infusion for 96 h (days 1–5 and 29–33), together with mitomycin C (10 mg/m^2^) given as bolus (days 1 and 29). A total of two concurrent cycles were planned for each patient.

### 2.7. Toxicity Evaluation and Clinical Assessment

Acute toxicity was scored following the Common Toxicity Criteria for Adverse Events scale v4.0 (CTCAEv4.0), including gastro-intestinal (GI), genitourinary (GU), dermatologic, hematologic, events and recorded within 90 days from the end of treatment [21]. During observation, patients were re-assessed with digital rectal examination, anoscopy, pelvic MRI and ^18^FDG-PET. Final response evaluation was carried out 26 weeks after the start of RT-CHT.

### 2.8. Hematologic Toxicity Evaluation

All patients underwent a weekly complete blood count. HT was graded according to CTCAEv4.0 grading system. Endpoints evaluated in the present analysis were white blood cell count (WBC), absolute neutrophil count (ANC), hemoglobin (Hb) and platelet (Plt) count nadirs after each chemotherapy cycle and the highest-grade toxicity for all blood cells. HT was defined as each hematologic event with a grade higher than 3.

### 2.9. Statistical Analysis

Discrete and continuous variables were summarized by frequencies and percentages and using standard measures of central tendency and dispersion. The time-to-event functions were estimated by the Kaplan–Meier product-limit method. Colostomy-free survival was (CFS) defined as the time between RT-CHT start and the date of colostomy, death, or last follow-up. Failure-free survival (FFS) was defined as the time between RT-CHT and the date of any treatment-failure either local, regional or distant. We explored whether clinical and/or dosimetric characteristics were different between patients experiencing major acute HT (G3–G4 events) and those who did not (G0–G2). For continuous variables, the normal distribution was assessed verifying the values of skewness and kurtosis. Mean (SD) scores and frequencies were used as descriptive analyses, in case of normal distribution. The ANOVA test was used to explore differences between continuous variables (age, ^ACT^PBM V_10_, ^ACT^PBM V_20_, ^ACT^LSBM V_40_, and ^ACT^LSBM mean dose), while χ2-test r Fisher exact testing were used to test differences between categorical variables (gender, tumor, and nodal stage). A *p*-value <0.05 was considered as statistically significant. All the analyses were carried out with MedCalc^®^ Statistical Software version 19.6.1.

## 3. Results

A total of 39 patients was finally included in this prospective phase II trial. Detailed patients’ characteristics can be found in Table 1. Mean age was 64 (range 29–81) and patients were mainly female (77%), HIV-negative (95%), with an anal canal primary tumor (90%), T2–T3 stage (74%), N1 nodal disease (51%), and global stage IIIC (33%). No patient underwent a preventive colostomy. Patients were mainly treated with a dual-arc VMAT approach (92%), up to a total dose to the primary tumor PTV of 54 Gy (92%) and to 45 Gy (92%) to the prophylactic volumes delivered with conventional fractionation. A total of 20 node positive patients, also received a simultaneous integrated boost to the macroscopic nodal disease mostly up to 50.4 Gy (60%). Mean radiotherapy duration time was 45 days. All but two patients were submitted to two cycles of CHT (95%) with dose reduction during treatment observed in 8% of patients (average dose reduction: 20% of the planned dose). Three patients had a temporary interruption of the radiotherapy course (average duration: 2 days). See Table 2 for details.

### 3.1. Acute Hematologic Toxicity and Dosimetric Outcomes

Mean value at baseline for WBC was 7.380/cm^3^ (SD:2.560), which dropped to a minimum of 3.380/cm^3^ (SD:1.290) one week after the end of RT-CHT (Week 7), reaching 5.600/cm^3^ (SD:2.470) at three months after the end of treatment. Absolute neutrophil count at baseline was 4.730/cm^3^ (SD:2.110), which dropped to a minimum of 2.510/cm^3^ (SD:1.210) at Week 7, reaching 3.740/cm^3^ (SD:1.760) at three months. Mean value at baseline for Plts was 264.000/cm^3^ (SD:58.000), which dropped to a minimum of 178.000/cm^3^ (SD:67.000) at Week 7, reaching 251.000/cm^3^ (SD:65.000) at three months after RT-CHT. For Hb, mean value at baseline was 13.1 g/dl (SD:1.2), which dropped to a minimum of 11.1 g/dl (SD:1.6) at Week 7, to reach 12.2 g/dl (SD:1.6) at three months. Figure 1 shows the weekly trend for the analyzed blood parameters during and after concurrent RT-CHT.

Major non-hematologic toxicities comprised G3 events for skin and genitalia in 18% and 3% of patients, respectively (Table 3).

Maximum detected acute HT comprised eight observed events of leukopenia ≥G3 (20%) and 10 events of neutropenia ≥G3 (25%). Four events of ≥G3 thrombocytopenia (11%) were detected and only one patient (3%) experienced G3 anemia (Table 3). Overall, 11 out 39 of treated patients (28%) experienced ≥G3 acute HT. That corresponds to 28 patients (72%) experiencing G0–G2 events with respect to acute HT, with an overall threshold set by the trial design at 21 patients (54%) at least.

Dosimetric parameters pertinent to both treatment volumes and standard organs at risk are shown in Table 4. Those relative to active BM are presented in Table 5.

No significant differences were found between patients experiencing G3–G4 acute HT and those who did not in terms of age, gender, tumor, and nodal stage and most of the dosimetric parameters employed during the optimization process (^ACT^PBM V_10_, ^ACT^PBM V_20_, and ^ACT^LSBM mean dose). A borderline statistically significant difference (*p* = 0.055) was observed in terms of ^ACT^LSBM V_40_. Specifically mean ^ACT^LSBM V_40_ was found to be 24.8% (SD: 7.8) for patients not experiencing major acute HT and 30.4% (SD: 9.8) for those in which G3-G4 acute HT was observed (Figure 2).

### 3.2. Oncological Outcomes

Median observation time for the whole cohort was 19.5 months (range 6–36). At the time of last follow up, all patients were alive with 36/39 (92.3%) being free of disease. Seven patients (17.9%) experienced treatment failure, with four having local relapse, two having regional nodal isolated recurrence, and one experiencing synchronous regional and systemic failure. The four patients with local relapse underwent salvage surgery consisting of abdomino-perineal resection according to Miles, while the two experiencing nodal failure within the inguinal groins were given bilateral inguinal lymphadenectomy. All six patients were salvaged with surgery and are with no evidence of disease at last follow up examination. Two-year CFS was 93.0% (95% CI: 74.6–98.2%), while two-year FFS was 83.7% (95% CI: 61.2–93.7%) (Figure 3 and Figure 4).

## 4. Discussion

Concurrent RT-CHT is the standard of care in patients affected with anal cancer, as demonstrated in the ACT-I and EORTC 22,861 trials [22,23]. Intensified chemotherapy regimens with continuous intravenous infusion of 5-fluorouracil 1000 mg/m^2^ on days 1–4 (or 750 mg/m^2^ on days 1–5) and days 29–33 and mitomycin C 12–15 mg/m^2^ on day 1 were employed in those trials [22,23]. Of notice, these European landmark trials have typically administered to patients 1 cycle of concurrent mitomycin C. Conversely, in the North American trials, namely the RTOG8704-ECOG1289 and RTOG 9811 trials, two cycles of mitomycin C at a dose of 10 mg/m^2^ were given [5,24]. The addition of chemotherapy to radiation in this clinical setting is responsible of improved clinical outcomes, but also increased toxicity. As an example, in the ACT-I trial, six deaths were attributed to chemotherapy and a higher rate of severe skin (50%) and GI (14%) toxicities was observed in the combination therapy arm [22]. In the setting of anal cancer patients submitted to combination therapy, acute HT can limit the possibility to deliver a full course of treatment with definitive intent, hampering the overall treatment intensity, due to the increased likelihood to develop major side effects including bleeding, infections and/or asthenia [4]. Due to its direct myelosuppressive effect, chemotherapy is considered the most important trigger for HT [4]. In the EORTC 22,861 trial, one treatment-related death was ascribed to sepsis [23]. Severe acute HT was observed in 26% of patients allocated to the standard arm of the ACT-II trial and a 3% rate of neutropenic sepsis was observed^1^. In the RTOG8704-ECOG1289, the rate of grade 4 and 5 HT was 20% with four patients (3%) experiencing treatment-related death [24]. In the standard arm of the RTOG 9811 trial, up to 61% of patients experienced grade 3–4 acute HT [5]. To limit acute HT, different strategies can be employed and some of them are targeted to the choice of the chemotherapy regimen. One option could be to limit the administration of mitomycin to only one cycle at the beginning of the RT-CHT course. As shown in the retrospective study by White et al., comparing anal cancer patients undergoing concurrent RT-CHT with either one or two cycles of mitomycin, patients receiving one cycle only had lower rates of clinically significant acute HT including neutropenia, with three treatment-related deaths observed in the cohort of patients receiving two cycles of mitomycin and due to neutropenic sepsis [25]. Conversely, loco-regional control and survival outcomes were found to be similar, regardless of the number of cycles administered [25]. Another approach that can be considered is the concurrent use of cisplatin instead of mitomycin, which was demonstrated to be well-tolerated and to yield to similar survival outcomes as mitomicin in the experimental arm of the ACT II trial [1]. Interestingly, the rate of acute grade 3–4 HT was observed to be 16% for cisplatin and 26% for mitomicin in the aforementioned study [1].

A different approach is to act on radiotherapy. Bone marrow has been demonstrated to be a key dose-limiting cell renewal tissue during wide-field irradiation [4]. This concept relies on the observation that BM stem cells showed exquisite radiosensitivity, since radiotherapy has a strong myelosuppressive effect, inducing both apoptosis and stromal damage, characterized by peculiar pathologic and radiographic modifications [4]. Hence, the implementation of selective radiotherapy approaches, taking advantage of modern technologies, may represent an option to limit the damage to hematopoietic precursors. Intensity modulated radiotherapy ‘per se’ may be an option, as demonstrated by the results of the RTOG 0529 trial, where the use of IMRT was able to reduce the rate of ≥G2 acute HT, compared to the rates observed in the RTOG 9811 study [6]. However, the same trial showed a rate of ≥G3 acute HT as high as 58%, which calls for a selective avoidance strategy targeting those structures with a hematopoietic function. The most important BM site with functional activity in the adult population is the pelvic region and lumbar vertebrae which comprise around 60% of the active BM. In particular, the osseous segments within the pelvis may contain up to 40% of the total functional BM [4,11]. Since the radiotherapy approach employed in anal cancer patients is made of extended-volume irradiation, the unintended dose received by the pelvic bones may be a contributing factor in determining HT in this setting. Interestingly, the extent of radiation-induced BM damage has been reported to be influenced by both total radiation dose and BM volume receiving irradiation [4].

Hence, we have decided to investigate, within a prospective phase II trial, the selective avoidance of pelvic active BM employing IMRT to decrease the acute HT profile in anal cancer patients undergoing concurrent RT-CHT. To identify active BM, we employed functional imaging and, in particular, ^18^FDG-PET which provides an individual mapping of BM distribution accounting for variations depending on gender and age and allows for a reduction in the BM volumes as compared to delineation approaches based on the use of the whole bony structures [11]. Active BM was then considered as an organ at risk to be used during IMRT optimization [12]. Dose constraints directed to active BM addressed both low dose to ^ACT^PBM and medium-high doses to ^ACT^LSBM. Low doses to PBM were shown to be correlated to decreased blood cell nadirs and increased likelihood to experience acute HT in anal cancer patients [7]. We hence chose ^ACT^PBM V_10_ and ^ACT^PBMV_20_ < 90% and <75%, respectively, as per the INTERTECC-2 trial, which investigated the role of BM-sparing IMRT in cervical cancer patients [26]. Another target to be selectively spared is LSBM, because a) the relative proportion of active BM within LSBM is high, b) the location of ^ACT^LSBM is central and in close proximity to high-dose volumes during radiotherapy [11,27]. Hence, we chose^ACT^LSBM-V_40_ < 41% and ^ACT^LSBM mean dose < 32 Gy, as cut-off points, as previously investigated in our studies [9,11,28]. Targeting both PBM and LSBM during the optimization process allowed us to minimize the interplay effect between low dose to PBM and the tolerance threshold of LSBM to RT, as demonstrated by our group [20].

The acute toxicity profile with respect to non-hematologic endpoints was generally mild, with limited major toxicities (≥G3), mostly observed within the skin (18%) and genitalia (3%). Interestingly no major GI event was recorded. With respect to the primary endpoint of the study, namely acute HT, 11 out of 39 patients (19%), treated with BM-sparing IMRT, experienced ≥ G3 events, with neutropenia (25%), leukopenia (20%) and thrombocytopenia (11%) as most frequent observations. A total of 28 out of 39 patients (72%) experienced G0–G2 acute HT, above the threshold set by the trial design at 54%. We were, hence, able to improve by 30% the rate of minor acute HT compared to the data obtained in the RTOG 0529 trial, above the threshold of 20% improvement expected while designing the study [6]. These results allowed us to reject the null hypothesis (no difference in acute HT between standard and BM-sparing IMRT) and to potentially fulfil the criteria to define BM-sparing IMRT as a ‘promising’ treatment for anal cancer patients undergoing concurrent CHT-RT with definitive intent to reduce the acute hematologic toxicity profile.

Given the small sample size, we were not able to perform a univariate/multivariate analysis. Nevertheless, we compared clinical characteristics and dosimetric parameters between patients experiencing major acute HT and those who did not. The only parameter with a borderline statistically significant difference was ^ACT^LSBM V_40_, which was higher for patients with ≥G3 acute HT. This trend suggests the possibility to even lower the limit below the 41% threshold that we set in our study, in order to further spare hematopoietic precursors located in the lumbar–sacral region.

## 5. Conclusions

This prospective phase II trial supports the feasibility and effectiveness of BM-sparing IMRT in reducing acute HT in anal cancer patients submitted to concurrent RT-CHT with definitive intent. This is obtained with no detrimental effect on tumor control and patient’s survival, since oncological outcomes were shown to be consistent with the available literature, considering the case mix of the present cohort comprising more than a half of locally advanced cases. Further investigations are nevertheless needed to increase the robustness of this finding and support the clinical dissemination of this approach.

## Figures and Tables

**Figure 1 jpm-11-00427-f001:**
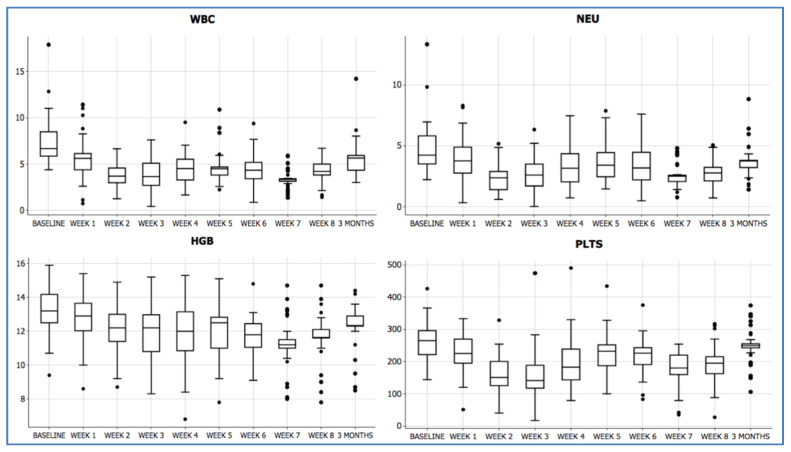
Acute hematologic toxicity during and after treatment.

**Figure 2 jpm-11-00427-f002:**
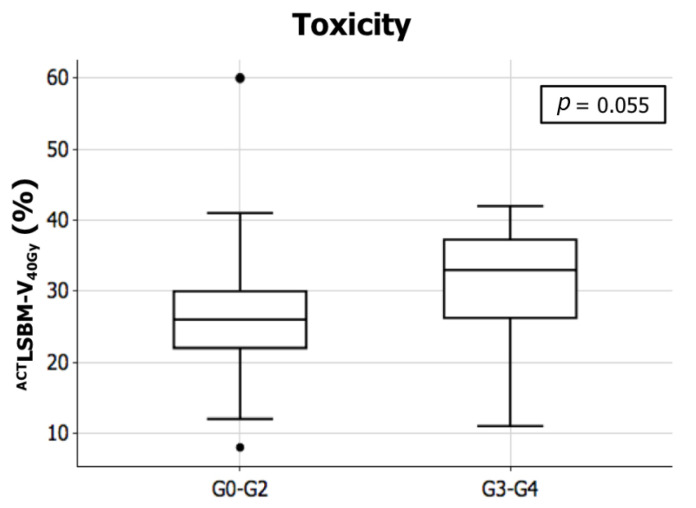
Mean values for the volume of active lumbar sacral bone marrow receiving 40 Gy (LSBM PET V_40_) in patients experiencing major acute hematologic toxicity (G3–G4) compared to those who did not (G0–G2).

**Figure 3 jpm-11-00427-f003:**
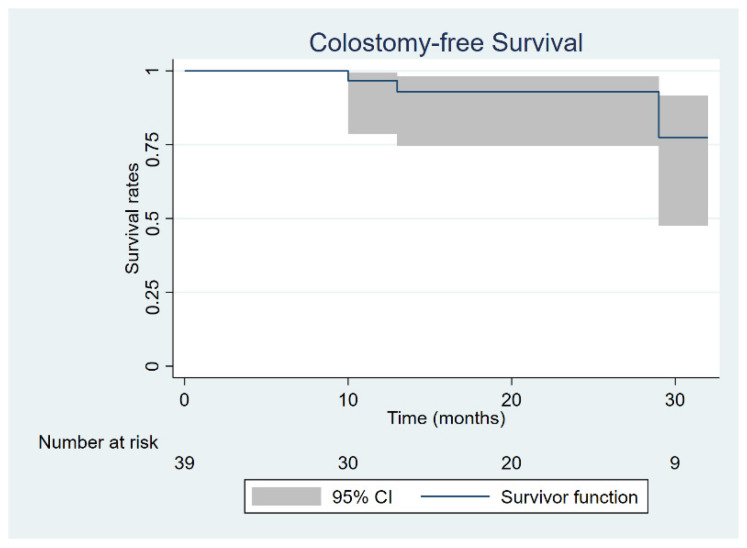
Colostomy-free survival.

**Figure 4 jpm-11-00427-f004:**
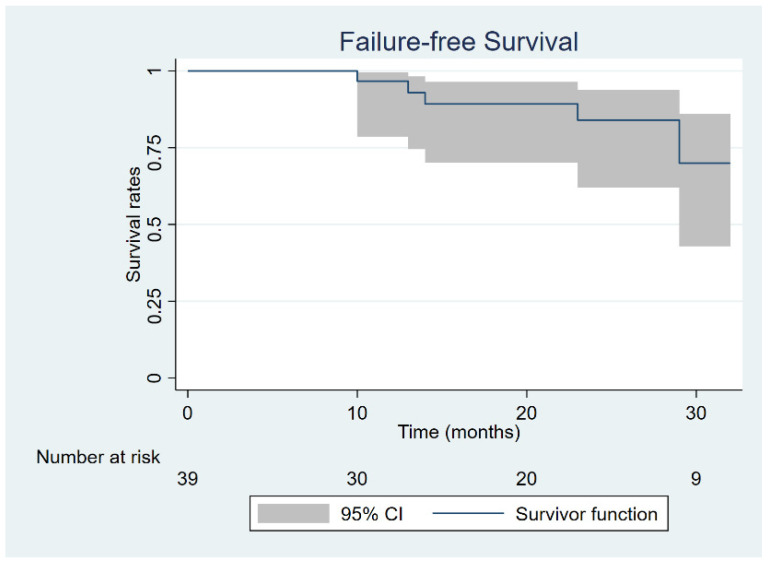
Failure-free survival.

**Table 1 jpm-11-00427-t001:** Patient and tumor characteristics.

Variable	N (%)
**Age**	
Mean	64
Range	29–81
**Gender**	
Female	30 (77)
Male	9 (23)
**HIV status**	
Positive	2 (5)
Negative	37(95)
**Primary tumor site**	
Anal canal	35 (90)
Anal margin	1 (2)
Both	3 (8)
**T-stage**	
T1	4 (10)
T2	15 (38)
T3	14 (36)
T4	6 (15)
**N-stage**	
N0	19 (49)
N1	20 (51)
**Global stage**	
I	3 (8)
IIA	12 (31)
IIB	4 (10)
IIIA	6 (15)
IIIB	1 (3)
IIIC	13 (33)
**Grading**	
G1	5 (13)
G2	10 (26)
G3	11 (28)
NA	13 (33)

**Table 2 jpm-11-00427-t002:** Treatment characteristics.

Variable	N (%)
**IMRT approach**	
Single-arc	3 (8)
Dual-arc	36 (92)
**PTV dose-tumor (Gy)**	
54 Gy/30 fractions	36 (92)
50.4 Gy/28 fractions	3 (8)
**PTV dose-positive nodes (Gy) 20 pts**	
54 Gy/30 fractions	8 (40)
50.4 Gy/30 fractions	12 (60)
**PTV dose-negative nodes (Gy)**	
45 Gy/30 fractions	36 (92)
42 Gy/30 fractions	3 (8)
**Chemotherapy**	
5-FU + MMC	39 (100)
**Cycles**	
1	2 (5)
2	37(95)
**Chemotherapy dose reduction**	
Yes	3 (8)
No	36 (92)
**RT duration (days)**	
Mean	45
Range	37–77

**Table 3 jpm-11-00427-t003:** Acute toxicity profile.

Acute Toxicity	G0	G1	G2	G3	G4
Skin	0 (0)	9 (23)	23 (59)	7 (18)	0 (0)
Gastrointestinal	7 (18)	13 (33)	19 (49)	0 (0)	0 (0)
Urinary	11 (28)	19 (49)	9 (23)	0 (0)	0 (0)
Genitalia	12 (31)	16 (41)	10 (26)	1 (3)	0 (0)
Anemia	22 (56)	9 (23)	7 (18)	1 (3)	0 (0)
Leukopenia	4 (10)	11 (28)	16 (41)	4 (10)	4 (10)
Neutropenia	14 (36)	7 (18)	8 (21)	6 (15)	4 (10)
Thrombocytopenia	26 (67)	8 (21)	1 (3)	3 (8)	1 (3)

**Table 4 jpm-11-00427-t004:** Dosimetric parameters for both target and organs at risk.

**PTV**			
		mean	SD
PTV-tumor	D_98_ (Gy)-50.4Gy	48.0	1.0
	D_2_ (Gy)-50.4Gy	53.0	2.0
	D_98_ (Gy)-54Gy	51.2	1.8
	D_2_ (Gy)-54Gy	57.1	1.2
	V_95_ (%)	95.5	3.6
	V_107_ (%)	1.6	1.5
PTV-elective volumes	D_98_ (Gy)-42Gy	48	0
	D_2_ (Gy)-42Gy	54	0
	D_98_ (Gy)-45Gy	50.2	2.44
	D_2_ (Gy)-45Gy	57.55	1.36
	V_95_ (%)	91.45	8.61
	V_107_ (%)	2.66	2.81
**OARs**			
		mean	SD
Bladder	V_30_ (%)	40.3	13.2
	V_40_ (%)	19.3	13.1
	V_50_ (%)	4.5	9.3
	D_2_(Gy)	47.3	4.8
	Mean dose (Gy)	27.8	4.5
Bowel	V_30_ (cc)	198.1	94.8
	V_35_ (cc)	145.6	88.1
	V_40_ (cc)	19.2	32.7
	V_45_ (cc)	9.3	36.3
	D_2_(Gy)	44.3	3.0
	Mean dose (Gy)	20.4	12.1
External genitalia	V_20_ (%)	40.8	25.5
	V_30_ (%)	29.5	22.6
	V_40_ (%)	15.3	18.4
	D_2_(Gy)	49.4	11.9
	Mean dose (Gy)	23.3	11.1
Femural heads	V_30_ (%)	10.8	8.3
	V_40_ (%)	2.4	4.5
	V_45_ (%)	0.8	3.4
	V_50_ (%)	0.5	3.2
	D_2_(Gy)	36.6	7.6
	Mean dose (Gy)	17.5	8.8

**Table 5 jpm-11-00427-t005:** Dosimetric parameters for pelvic active bone marrow and its subsites.

Structure				Structure			
	Parameter	Mean	SD		Parameter	Mean	SD
**^ACT^PBM**	D_mean_(Gy)	23.6	4.1	**^ACT^IBM**	D_mean_(Gy)	19.9	4.13
	V_5_	94.1	8.1		V_5_	93.8	7.8
	V_10_	81.8	12.1		V_10_	77.1	12.7
	V_15_	68.1	15.1		V_15_	66.4	15.9
	V_20_	55.6	15.1		V_20_	44.8	16.3
	V_30_	33.9	11.1		V_30_	20.2	11.8
	V_40_	15.4	6.4		V_40_	6.4	5.7
	V_45_	4.1	3.4		V_45_	1.3	2.6
	V_50_	0.5	1.1		V_50_	0.1	0.4
**^ACT^LSBM**	D_mean_(Gy)	29.7	11.8	**^ACT^LPBM**	D_mean_ (Gy)	24.5	5.8
	V_5_	95.22	7.8		V_5_	96.6	8.8
	V_10_	89.2	11.3		V_10_	85.9	15.9
	V_15_	81.5	13.9		V_15_	70.5	21.2
	V_20_	72.8	14.2		V_20_	57.3	21.1
	V_30_	52.8	12.7		V_30_	35.0	16.2
	V_40_	27.4	10.2		V_40_	16.1	9.6
	V_45_	7.1	5.6		V_45_	5.1	4.9
	V_50_	0.9	2.1		V_50_	0.7	1.6

## Data Availability

Data related to the present study can be found at the corresponding author upon request.

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
