# Peer review of "Concurrent Chemoradiation in Anal Cancer Patients Delivered with Bone Marrow-Sparing IMRT: Final Results of a Prospective Phase II Trial"

_jpm, 2021, doi:10.3390/jpm11050427_

Round 1

Reviewer 1 Report

Thank you for inviting me to review this interesting manuscript.

The authors used a bone-marrow sparing IRMT technique to reduce severe hematologic toxicity in patients treated for anal cancer.

It is small patient population (39 patients) prohibiting the use of univariate or multivariate analysis. Instead the authors used an interesting statistical approach.

Comments

The standard protocol were two cycles of chemotherapy. Apparently two patients only received one cycle. I recommend to exclude these 2 patients because chemotherapy has a major influence on HT, the main outcome paramter of the study.

Please define how much of dose reduction was necessary in 3 patients.

In how many patients an interruption or stop of radiotherapy or chemotherapy was necessary?

It would be of interest to have a CT image or scheme of how the bone marrow segementation was done.

Minor comments:

Headings of Table 1-4 are confounded.

Please explain all abbreviations used in tables.

Last page of results line 229 and 230: Replace Gy by %

Author Response

We would like to thanks the reviewers for their thorough comments and helpful suggestions which contributed to improve our manuscript.  We hereby provide a point-by-point reply to their comments.

Reviewer 1

Thank you for inviting me to review this interesting manuscript.

The authors used a bone-marrow sparing IRMT technique to reduce severe hematologic toxicity in patients treated for anal cancer.

It is small patient population (39 patients) prohibiting the use of univariate or multivariate analysis. Instead the authors used an interesting statistical approach.

Thank you for your comments.

Comments

The standard protocol were two cycles of chemotherapy. Apparently two patients only received one cycle. I recommend to exclude these 2 patients because chemotherapy has a major influence on HT, the main outcome paramter of the study.

Thanks for your comment. This is a prospective phase II trial with a predetermined sample size calculation and hence the analysis has been done on the whole cohort of patients. The 2 patients who only received one cycle of chemotherapy were amongst those who experienced major hematologic toxicity (this is the reason why they did not receive the second cycle). Their removal from the analysis would artificially improve the clinical results of the study and hence we would discourage it. We would rather maintain the analysis on the whole set of patients.

Please define how much of dose reduction was necessary in 3 patients.

Thank you for the comments. The average dose reduction was  20% of the planned dose. This sentence has been added in the text within the results section.

In how many patients an interruption or stop of radiotherapy or chemotherapy was necessary?

Three patients had a temporary interruption of the radiotherapy course (average duration: 2 days). This sentence has been added in the text within the Results section.

It would be of interest to have a CT image or scheme of how the bone marrow segementation was done.

Details on the segmentation process have already been the object of different publications by our group. We added a sentence addressing the publications of interest.

  1. Franco, P. et al. Dosimetric predictors of acute hematologic toxicity during concurrent intensity-modulated radiotherapy and chemotherapy for anal cancer. Transl. Oncol. 19, 67-75 (2017), doi: 10.1007/s12094-016-1504-2. 
  2. Franco, P. et al. Dose to specific subregions of pelvic bone marrow defined with FDG-PET as a predictor of hematologic nadirs during concomitant chemoradiation in anal cancer patients. Oncol. 33:72 (2016). doi: 10.1007/s12032-016-0789-x.
  3. Arcadipane, F. et al. Bone marrow-sparing IMRT in anal cancer patients undergoing concurrent chemo-radiation: results of the first phase of a prospective phase II trial. Cancers (Basel) 12:3306 (2020). doi: 10.3390/cancers12113306.

Minor comments:

Headings of Table 1-4 are confounded.

Thanks for noticing. Headings were corrected.

Please explain all abbreviations used in tables.

Thanks for noticing. All abbreviations are now explained.

Last page of results line 229 and 230: Replace Gy by %

Thank you. Corrected.

Reviewer 2 Report

Concurrent chemoradiation in anal cancer patients delivered 2 with bone marrow-sparing IMRT: final results of a prospective phase II trial.

This is an interesting report of reduced toxicity using targeted IMRT.  The study accrued the full planned number of subjects (N=39), and toxicity outcomes are reported.

An interim analysis of 21 subjects was published in "Cancers" (MDPI) in 2020 (reference 12), and the findings from the full N=39 are broadly consistent with the interim N=21.  Reference 12 was submitted for publication in October 2020, the very month in which accrual was completed.  It is very puzzling that the authors chose to submit reference 12 (N=21) for publication when surely they must have known that accrual was already very close to (or at) the target of N=39.

It is also notable that no efficacy outcome (e.g. survival) data are provided in the manuscript, which leaves open the possibility that reduced toxicity might have been achieved at the expense of reduced efficacy.

I would recommend that the authors resubmit the manuscript with the inclusion of efficacy (e.g. progression or survival) data, in order to distinguish it from reference 12.

Author Response

Reviewer 2

Concurrent chemoradiation in anal cancer patients delivered 2 with bone marrow-sparing IMRT: final results of a prospective phase II trial.

This is an interesting report of reduced toxicity using targeted IMRT.  The study accrued the full planned number of subjects (N=39), and toxicity outcomes are reported.

Thank you for your comments.

An interim analysis of 21 subjects was published in "Cancers" (MDPI) in 2020 (reference 12), and the findings from the full N=39 are broadly consistent with the interim N=21.  Reference 12 was submitted for publication in October 2020, the very month in which accrual was completed.  It is very puzzling that the authors chose to submit reference 12 (N=21) for publication when surely they must have known that accrual was already very close to (or at) the target of N=39.

Thanks for your comment but, honestly it is just a coincidence. The timeline was only determined by the time available for the writing author (Pierfrancesco Franco) to complete the analysis and the draft of the manuscript. There is no hidden reason for the observation pointed out by the reviewer.

It is also notable that no efficacy outcome (e.g. survival) data are provided in the manuscript, which leaves open the possibility that reduced toxicity might have been achieved at the expense of reduced efficacy.

I would recommend that the authors resubmit the manuscript with the inclusion of efficacy (e.g. progression or survival) data, in order to distinguish it from reference 12.

Thank you. This a good comment. A survival analysis has been added, with the number of failure events and their pattern, together with data on colostomy-free and failure-free survival.

Round 2

Reviewer 2 Report

The authors have addressed the reviewer's comments